# Cell-Membrane Biomimetic Indocyanine Green Liposomes for Phototheranostics of Echinococcosis

**DOI:** 10.3390/bios12050311

**Published:** 2022-05-09

**Authors:** Xinxin Xiong, Jun Li, Duyang Gao, Zonghai Sheng, Hairong Zheng, Wenya Liu

**Affiliations:** 1The First Affiliated Hospital of Xinjiang Medical University, Urumqi 830011, China; 13095025355@163.com (X.X.); lijun@xjmu.edu.cn (J.L.); 2Paul C. Lauterbur Research Center for Biomedical Imaging, Institute of Biomedical and Health Engineering, Shenzhen Institute of Advanced Technology, Chinese Academy of Sciences, Shenzhen 518055, China; dy.gao@siat.ac.cn (D.G.); zh.sheng@siat.ac.cn (Z.S.); hr.zheng@siat.ac.cn (H.Z.)

**Keywords:** fluorescence imaging, photothermal therapy, indocyanine green, cell-membrane biomimetic engineering, echinococcosis

## Abstract

Echinococcosis is an important zoonotic infectious disease that seriously affects human health. Conventional diagnosis of echinococcosis relies on the application of large-scale imaging equipment, which is difficult to promote in remote areas. Meanwhile, surgery and chemotherapy for echinococcosis can cause serious trauma and side effects. Thus, the development of simple and effective treatment strategies is of great significance for the diagnosis and treatment of echinococcosis. Herein, we designed a phototheranostic system utilizing neutrophil-membrane-camouflaged indocyanine green liposomes (Lipo-ICG) for active targeting the near-infrared fluorescence diagnosis and photothermal therapy of echinococcosis. The biomimetic Lipo-ICG exhibits a remarkable photo-to-heat converting performance and desirable active-targeting features by the inflammatory chemotaxis of the neutrophil membrane. In-vitro and in-vivo studies reveal that biomimetic Lipo-ICG with high biocompatibility can achieve in-vivo near-infrared fluorescence imaging and phototherapy of echinococcosis in mouse models. Our research is the first to apply bionanomaterials to the phototherapy of echinococcosis, which provides a new standard for the convenient and noninvasive detection and treatment of zoonotic diseases.

## 1. Introduction

Echinococcosis, also known as hydatid disease, is a serious infectious disease caused by the larval stages of cestodes of the genus echinococcus [1,2,3]. Echinococcus is an important zoonotic infectious disease that can be widely spread among host dogs, intermediate host sheep and cattle, and humans, and which is seriously affecting human health and livestock production [4,5,6]. Currently, conventional medical imaging techniques, including chest radiography [7], X-ray computed tomography [8], magnetic resonance imaging [9], and ultrasound imaging [10], have been applied for the clinical diagnosis of echinococcosis. Surgery and medication are adopted as the main methods to treat echinococcosis [11,12,13]. Nevertheless, it is difficult to perform advanced imaging procedures to diagnose and evaluate therapeutic efficacy in remote pastoral areas [14]. Furthermore, surgical trauma and serious side effects of medications also hinder their widespread applications [15]. Therefore, there is an urgent need to design convenient, noninvasive and efficient methods for the theranostics of echinococcosis.

Phototheranostics is a hybrid technology that utilizes small molecule probes or nanoprobes for optical molecular imaging-guided phototherapy for a variety of diseases, including cancer [16,17,18], bacterial infections [19], and neurodegenerative diseases [20]. Compared with traditional medical imaging and treatment methods, it has unique advantages such as high sensitivity, good selectivity, and non-invasiveness [21,22,23,24,25]. Currently, a variety of phototheranostic agents, such as indocyanine green (ICG) [26], polymer dots [27], carbon nanotubes [28], graphene [29], gold nanorods [30], etc. [31], have been developed for high-performance phototheranostics. Among these materials, only ICG is clinically used as the U.S. Food and Drug Administration-approved agent [32,33,34], offering great potential for phototheranostics of zoonotic diseases. Nevertheless, ICG suffers from short blood half-life, low stability, and a non-active targeting ability, which hinders its application in hydatid disease [35,36,37,38].

Herein, we report the use of neutrophil-membrane camouflaged ICG liposomes (Neu-lipo-ICG) for active targeting the phototheranostics of echinococcosis in a mouse model. Liver tissues infected with hydatid disease may produce severe inflammation, which accumulates a large number of neutrophils due to their inflammatory chemotaxis [35]. Inspired by unique characteristics, we prepared Neu-lipo-ICG using neutrophil-membrane camouflage technology to enhance its active targeting for the precise photodiagnosis and phototherapy of echinococcosis. Neu-lipo-ICG shows bright near-infrared fluorescence and a good photothermal performance. Importantly, it enables the performance of in-vivo fluorescence imaging to visualize echinococcosis lesions with a high signal-to-background ratio, and photothermal therapy under near-infrared laser irradiation. Our results provide a novel paradigm for the accurate diagnosis and effective treatment of echinococcosis.

## 2. Materials and Methods

First, 1,2-dioleoyl-sn-glycero-3-phosphocholine (DOPC) and 1,2-distearoyl-sn-glycero-3-phosphoethanolamine-N-[amino(polyethylene glycol)-2000] (DSPE-PEG_2000_) were purchased from Avanti (Alabaster, Birmingham, AL, USA). ICG was obtained from Sigma-Aldrich (St Louis, MO, USA). Fetal bovine serum (FBS) and Trypsin ethylenediaminetetraacetic acid (EDTA) were from Gibco Life Technologies (Grand Island, NY, USA).


**Preparation and characterization of Neu-lipo-ICG**


Lipo-ICG was prepared through a thin-film strategy. Briefly, DOPC and DSPE-PEG_2000_ with a molar ratio of 95/5 were dissolved in the CHCl_3_ and formed a thin film by being evaporated with nitrogen and dried in a vacuum for 4 h. The dry film was then rehydrated by adding PBS containing an adequate amount of ICG. The mixture was further subjected to freeze–thaw for 5 cycles using liquid nitrogen and 65 °C water bath, which was further extruded through a polycarbonate membrane with a pore diameter of 100 nm 15 times.

The improved Percoll gradient method was used to isolate peripheral blood neutrophils from fresh whole blood of mice. Briefly, after centrifugation purification and re-suspension in phosphate-buffered saline (PBS) containing ethylenediaminetetraacetic acid (EDTA), samples were carefully placed on a three-layer gradient in PBS diluted with 78%, 69%, and 52% Percoll. Neutrophils were collected from the 69–78% interface and the upper 78% layer by centrifugation at 800× *g* at 21 °C for 30 min in a test tube. 

To obtain neutrophils, the sample was gently suspended in a cold, hypotonic solution buffer with a cocktail of protease inhibitors. After 15 min of incubation in an ice bath, homogenization was performed 50 times with a Dounce homogenizer and a tightly mounted pestle. The homogenate was centrifuged at 700× *g* at 4 °C for 10 min to remove unbroken cells and nuclei. The neutrophil membrane was collected and centrifuged at 14,000× *g* for 30 min at 4 °C.

Neu-lipo-ICG was prepared with continuous extrusion and freezing treatments of Lipo-ICG and neutrophil membrane derived from activated neutrophils in mouse peripheral blood.

The purified Lipo-ICG and Neu-lipo-ICG solution were obtained by removal of the free ICG molecules through dialyzing (7 KD, 48 h). The encapsulation efficiency of the Lipo-ICG and Neu-lipo-ICG was further measured by the equation:$$Encapsulation\;efficiency\;\left(\%\right)=\frac{Mass\;of\;ICG\;in\;product}{Total\;mass\;of\;ICG}\times 100$$


**Mouse models infected with alveolar echinococcosis**


The animal experiments were approved by the Ethics Committee of Xinjiang Medical University. C57 mice (female, 20 ± 5 g, age of 8–10 weeks) were anesthetized with 10% chloral hydrate solution by intraperitoneal injection. We further injected protoscoleces (200–300 μL) into liver tissue through a routine procedure. Two months after these procedures, the mouse models were evaluated by ultrasound and magnetic resonance imaging.


**Separating protoscoleces from liver tissue in mouse models**


The mice infected with alveolus echinococcosis were euthanized, and their liver tissue was taken out by laparotomy. The obtained samples were cut into pieces and ground to form tissue homogenates. After filtration to remove necrotic tissue and erythrocytes, highly active protoscoleces were obtained with a concentration of 20% (2.0 × 10^4^ protoscoleces mL^−1^).


**In-vivo near-infrared fluorescence imaging**


Neu-lipo-ICG or Lipo-ICG (dose = 0.5 mg Kg^−1^) was administrated into mice infected with alveolus echinococcosis through tail-vein injection, and near-infrared fluorescent signals were detected at various time points (1, 3, 6, 12 and 24 h) using the commercial fluorescence imaging system (IVIS Spectrum, PerkinElmer, Waltham, MA, USA).


**In-vivo photothermal therapy**


The mice models of echinococcosis were intravenously injected with Lipo-ICG and Neu-lipo-ICG (dose = 0.5 mg Kg^−1^), respectively. After 24 h post-injection, near-infrared laser (808 nm) irradiation in liver tissue was performed for 10 min (0.6 W cm^−2^), and temperature variation was monitored by an infrared thermal camera (FLIR A300, Wilsonville, OR, USA). The photothermal efficacy was assessed by ultrasound (ACUSON Sequoia512, 8–12 MHz probes) and magnetic resonance imaging (GE3.0T Signa EXCITE T1WI).

## 3. Results

### 3.1. Synthesis and Characterization of Neu-Lipo-ICG

Neu-lipo-ICG was fabricated via continuous extrusion and freezing treatments of the Lipo-ICG and neutrophil membrane derived from activated neutrophils in mouse blood (Figure 1a). After the mechanical process, we conducted extensive physical and optical characterization of Neu-lipo-ICG. Transmission electron microscopy (TEM) images of negatively stained samples showed that Lipo-ICG and Neu-lipo-ICG exhibited an approximately spherical morphology with uniform size (Figure 1b,c). After the cell membrane camouflaging treatment, the particle size of Lipo-ICG decreased from ~128 nm to ~119 nm (Figure 1d); meanwhile, a coronal structure appeared in the shell layers of Lipo-ICG. It suggested that the neutrophil membrane was successfully modified onto the surface of Lipo-ICG by the extrusion and freezing treatments. The polydispersity index (PDI) of the Lipo-ICG and Neu-lipo-ICG was 0.126 and 0.098, respectively, indicating the nanoparticles possessed uniform size. Subsequently, the optical features of Neu-lipo-ICG were measured with absorption and emission spectra (Figure 1e,f). The results verified no obvious changes in the absorption and fluorescent emission peaks of Neu-lipo-ICG, correlating well with Lipo-ICG. The storage stabilities of Neu-lipo-ICG and the free ICG were evaluated by recording the fluorescence intensity under ambinet condition (25 °C, dark), respectively. The obtained Neu-lipo-ICG showed a higher storage stability as compared with free ICG under the same conditions (Figure 1g). In addition, the encapsulation efficiency of Lipo-ICG and Neu-Lipo-ICG were calculated to be 84.2% and 83.5%, respectively. Consequently, as revealed by the optical measurements, we speculated that Neu-lipo-ICG was competent for the fluorescence imaging of echinococcosis.

### 3.2. In-Vitro Photothermal Performance of Neu-Lipo-ICG

Inspired by strong optical absorption in the near-infrared window, we investigated the photothermal performance of Neu-lipo-ICG under 808 nm laser exposure. There was a notable temperature increase in Neu-lipo-ICG at different concentrations, ranging from 0 µg mL^−1^ to 50 µg mL^−1^ (Figure 2a). The temperature of the Neu-lipo-ICG solution could reach 45°C for hyperthermia under 808 nm laser irradiation (0.6 W cm^−2^, 50 µg mL^−1^) for 5 min. Nevertheless, a negligible temperature increment was detected for a phosphate buffer solution (PBS) under the same experimental conditions. Additionally, the temperature of the Neu-lipo-ICG solution increased with laser density elevation from 0.4 W cm^−2^ to 0.6 W cm^−2^ (Figure 2b). The dose-dependent, irradiation-duration-dependent and laser-density-dependent features enabled Neu-lipo-ICG to be an efficient photo-to-heat converting agent for photothermal therapy. Given the favorable photothermal performance of Neu-lipo-ICG, we then evaluated its photothermal-treatment efficacy against hydatid larva separated from the liver tissue in mouse models. Morphological changes in hydatids were observed by a bright-field microscope. Neu-lipo-ICG did not induce morphological changes in hydatids after incubation for 2 h (Figure 2c), indicating its good biocompatibility. In contrast, hydatids incubated with Neu-lipo-ICG were thermally destroyed under near-infrared laser exposure (0.6 W cm^−2^, 10 min) (Figure 2d), indicating a reduced size, missing surface hooks, and destroyed internal structures. The above results suggested a prominent photothermal effect of Neu-lipo-ICG that promoted hydatids’ death.

### 3.3. In-Vivo Biocompability of Neu-Lipo-ICG

Neu-lipo-ICG was prepared using the U.S. Food and Drug Administration-approved biomaterials (ICG, DSPE-PEG_2000_ and DOPC) with high biocompatibility. There was no evidence that Neu-lipo-ICG was not toxic in vivo. Thus, prior to in-vivo imaging, the toxicity of Neu-lipo-ICG was investigated to assess its biosafety for potential applications in clinics. Significantly, no evident changes in the blood indexes between the physiological saline-treated group (control) and Neu-lipo-ICG-treated group (low dose: 0.5 mg Kg^−1^ ICG, high dose: 2.0 mg Kg^−1^ ICG) were detected 24 h post-injection treatment (Figure 3a–i). Moreover, the results of the hematological and histological analysis of biomimetic ICG liposomes in mice are shown in Figure 3j,k. It shows that the hematological parameters of biomimetic ICG liposomes with a low dose and high dose did not change significantly compared with the control group. Meanwhile, the staining analysis of liver, spleen, lung and kidney tissues indicated no bleeding, inflammation or tissue necrosis. The results demonstrated biomimetic ICG liposomes have good biocompatibility in vivo.

### 3.4. In-Vivo Near-Infrared Fluorescence Imaging of Neu-Lipo-ICG

Subsequently, the in-vivo near-infrared fluorescence of Neu-lipo-ICG was investigated in mouse models using a highly sensitive imaging system (Figure 4a). As illustrated in Figure 4b, the fluorescence signals in liver regions were recorded after different time intervals post-injection. The sustained strong fluorescence signals in the Neu-lipo-ICG-treated group were observed over time, which were higher than that of Lipo-ICG-treated group. In comparison, the conventional Lipo-ICG-treated group showed inefficient accumulation in the liver infected tissue after 24 h post-treatment. Quantitative results further manifested that the fluorescence signal in the Neu-lipo-ICG group was 4.2-times higher than that of the Lipo-ICG-treated group 24 h post-injection (Figure 4c). These results revealed that Neu-lipo-ICG enabled an active-targeting performance in hydatid infection sites based on the inflammatory chemotaxis of neutrophil membrane proteins, and produced bright near-infrared fluorescence signals. It was noteworthy that biocompatible Neu-lipo-ICG could be completely degraded in liver tissue after performing imaging tasks, which was beneficial for clinical translation.

### 3.5. In-Vivo Photothermal Treatment of Neu-Lipo-ICG

Encouraged by the remarkable therapeutic effect in vitro and favorable active-targeting performance of Neu-lipo-ICG in vivo, its in-vivo photothermal-treatment efficacy was investigated in mouse models (Figure 5a). Mice infected with hydatidosis were randomly divided into three groups (n = 5), including group I: control, group II: laser irradiation, and group III: Neu-lipo-ICG + laser irradiation. The liver region was irradiated by an 808 nm laser (power density: 0.6 W cm^−1^) for 10 min at 24 h post-injection (dose: 0.5 mg Kg^−1^). The real-time temperature of the liver region was recorded by a thermal-imaging instrument during the photothermal treatment process. For mice intravenously administrated with Neu-lipo-ICG, the temperature at the liver sites rapidly increased by 15.4 °C during the first 3 min and then remained at 46.7 °C under near-infrared laser irradiation (Figure 5b). In comparison, the laser-irradiation-only group (group II) showed a slight temperature increase and reached 40 °C, which is lower than the hyperthermia threshold (42.0 °C), verifying its high safety. The lesion volumes in the liver tissues receiving different treatments were evaluated using ultrasound imaging and magnetic resonance imaging (Figure 5c), respectively. Neu-lipo-ICG-mediated photothermal therapy achieved significant alleviation of the lesion volume during the detected period of 7 days, validating that Neu-lipo-ICG efficiently killed hydatids under near-infrared laser irradiation. In contrast, laser irradiation alone or Neu-lipo-ICG administration without irradiation exhibited no effect on the hydatid infect suppression. These findings confirmed that Neu-lipo-ICG could be used as a high-performance photothermal agent with excellent biocompatibility for the treatment of echinococcosis.

## 4. Conclusions

In this study, we demonstrated that Neu-lipo-ICG is the ideal phototheranostic agent for near-infrared fluorescence-imaging-guided photothermal therapy of echinococcosis. The as-obtained Neu-lipo-ICG exhibited good optical and photothermal properties. In comparison with Lipo-ICG, the neutrophil-membrane-coating technology revealed a specific targeting ability for hydatid infected tissue through inflammatory chemotactic effects. Neu-lipo-ICG demonstrated a longer residence time in mouse models infected with alveolus echinococcosis than that of Lipo-ICG-treated group, due to its active-targeting ability. In addition, both in-vitro and in-vivo investigations manifested the Neu-lipo-ICG for the sensitive diagnosis and efficient therapy of echinococcosis in mouse models. It is noteworthy that this is the first prospective study using biomimetic liposomes for the diagnosis and treatment of zoonotic diseases, providing a promising method to explore functional nanomaterials for the rapid and efficient phototheragnostic of infectious diseases in the underdeveloped area.

## Figures and Tables

**Figure 1 biosensors-12-00311-f001:**
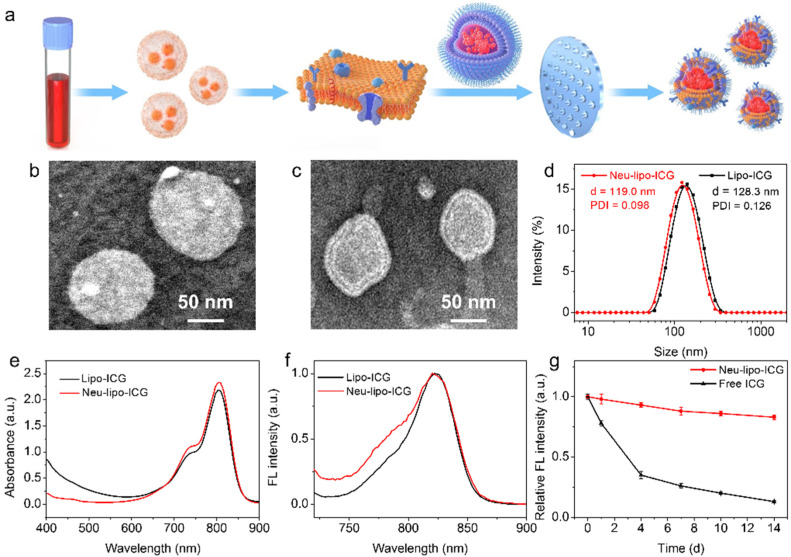
Synthesis of Neu-lipo-ICG. (**a**) Schematic representation of neutrophil membrane camouflaging Lipo-ICG. TEM images of Lipo-ICG (**b**) and Neu-lipo-ICG (**c**). (**d**) DLS measurements of Lipo-ICG and Neu-lipo-ICG. Absorption spectra (**e**) and emission fluorescence spectra (**f**) of Lipo-ICG and Neu-lipo-ICG (excitation: 710 nm). (**g**) Fluorescence varies of Neu-lipo-ICG and free ICG over storage (25 °C, dark).

**Figure 2 biosensors-12-00311-f002:**
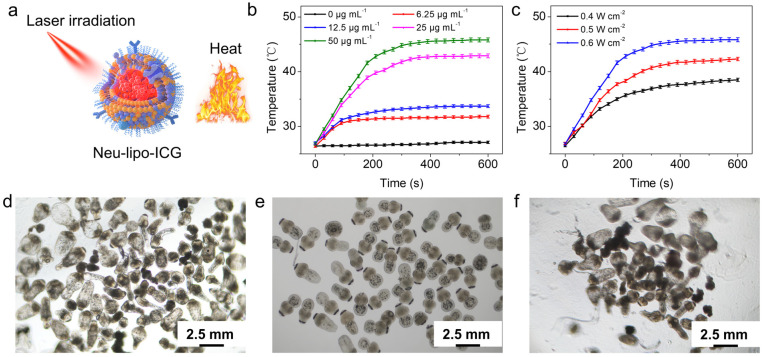
In-vitro photothermal-heating performance of Neu-lipo-ICG. (**a**) Schematic illustration of the photothermal therapy. (**b**) Photothermal-heating curves of Neu-lipo-ICG aqueous solution with different concentrations under 808 nm near-infrared laser irradiation (0.6 W cm^−2^). (**c**) Laser irradiation density-dependant photothermal-heating curves of Neu-lipo-ICG with a concentration of 50 µg mL^−1^. (**d**) Bright-field microscope images of Neu-lipo-ICG-treated protoscoleces, (**e**) laser-treated protoscoleces, and (**f**) Neu-lipo-ICG-laser-treated protoscoleces.

**Figure 3 biosensors-12-00311-f003:**
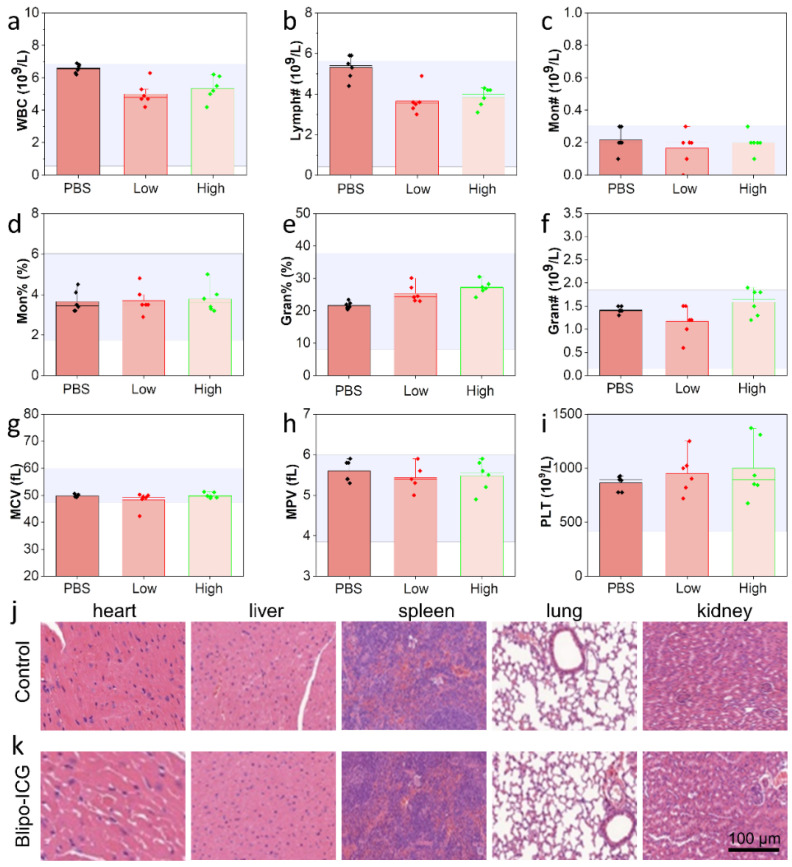
In-vivo toxicity evaluation of Neu-lipo-ICG. (**a**–**i**) Blood test parameters, including WBC (white blood cell), Lymph^#^ (lymphocyte number), Mon^#^ (monocyte number), Mon (monocyte), Gran (granulocytes), Gran^#^ (granulocytes number), MCV (mean corpuscular volume), MPV (mean platelet volume) and PLT (platelet) of Balb/c mice after various treatments (n = 5, low dose: low dose: 0.5 mg Kg^−1^ ICG, high dose: 2.0 mg Kg^−1^ ICG, 24 h). (**j**,**k**) H&E stained images were acquired from the major organs (heart, liver, spleen, lung, and kidney) of physiological saline-treated and Neu-lipo-ICG-treated groups at 24 h post-injection.

**Figure 4 biosensors-12-00311-f004:**
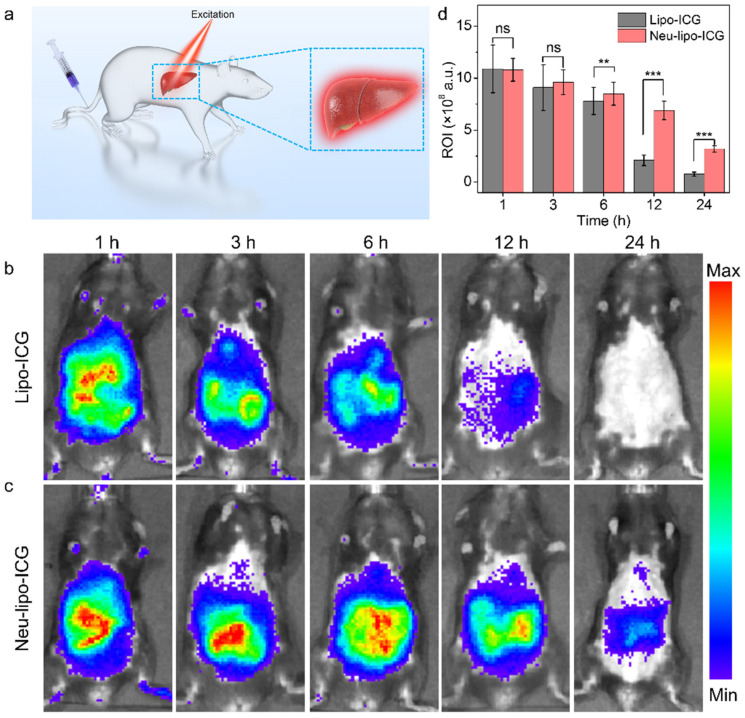
In-vivo near-infrared fluorescence imaging of Neu-lipo-ICG in mouse models infected with alveolus echinococcosis. (**a**) Scheme of in-vivo imaging by Neu-lipo-ICG. (**b**) Near-infrared fluorescence imaging of Lipo-ICG and (**c**) Neu-lipo-ICG at varied time points. (**d**) Fluorescence intensity of a region of interest (ROI) after intravenous injection of Neu-lipo-ICG (0.5 mg Kg^−1^) at various time points. Statistical analysis was performed using a student’s *t*-test, with *** indicating *p* < 0.005, ** indicating *p* < 0.05, ns indicating no significance.

**Figure 5 biosensors-12-00311-f005:**
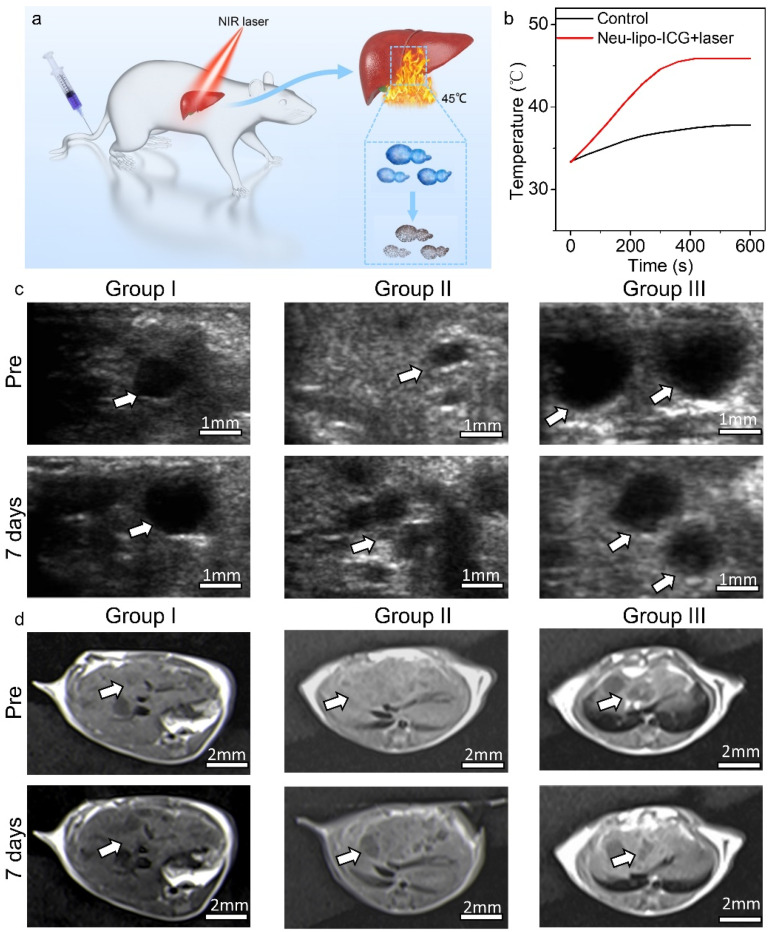
In-vivo photothermal therapy of Neu-lipo-ICG in mouse models infected with alveolus echinococcosis. (**a**) Scheme of in-vivo photothermal therapy by Neu-lipo-ICG. (**b**) Photothermal heating curves of the control group and Neu-lipo-ICG laser-treated group. (**c**) Ultrasound imaging and (**d**) magnetic resonance imaging of liver tissue of the echinococcosis-infected mice before and 7 days post-treatment.

## Data Availability

Not applicable.

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
