# Peer review of "Cell-Membrane Biomimetic Indocyanine Green Liposomes for Phototheranostics of Echinococcosis"

_biosensors, 2022, doi:10.3390/bios12050311_

Round 1

Reviewer 1 Report

n this manuscript by Xiong et al, the authors developed a neutrophil membrane encapsulated indocyanine green nanoparticle (Neu-Lipo-ICG). In vivo experiments using mouse models showed that the particle can be used for near-infrared imaging and photothermal therapy. The work can be interesting for researchers in the field of nanomedicine. However, the following questions need to be addressed before this manuscript can be accepted for publication.

  1. On Page 2, line 79-82, more details of material preparation should be given. For example, the cut-off size and time of dialysis should be provided. A detailed description of the extrusion and freezing procedure should be included. Also, how the neutrophil membrane is derived should be covered here.
  2. In section 3.1, ICG loading should be characterized quantitatively (eg, weight percent)
  3. On Page 4, line 124, the authors mentioned that the stability was compared "under the same conditions". The condition should be described in more detail (temperature, dark/light). Also, the following experiment is suggested to demonstrate the stability. ICG, Lipo-ICG and Neu-Lipo-ICG should be exposed under 808 nm laser over a period of time, and fluorescence intensity over time should be recorded.
  4. On Page 4, please double-check the standard deviation in Figure 1d is correct. It seems to be too small.
  5. In Section 3.2, the following control experiment should be included. Hydatids should be incubated without Neu-Lipo-ICG, but under near-infrared light. This can be used to exclude the possibility that light itself has an effect.
  6. In Section 3.3, it is suggested that immunogenicity assays (such as evaluation of cytokines release) should be included since the membrane is derived from cells.
  7. In Section 3.4, imaging by free ICG should also be included as a control group.
  8. In Section 3.5, histology analysis of liver tissue should be performed to make sure the treatment does not harm healthy tissues.
  9. The weight change of the mice should be recorded during treatment.

Reviewer 2 Report

Hi. Nice and original paper. I have reviewed the manuscript in MSWord format using track changes and comments. Please respond by accepting or rejecting the changes as appropriate. You need to respond to the comments, especially where you differ with me.

Reviewer 3 Report

Xiong et.al. designed neutrophil membrane camouflaging Lipo-ICG for active-targeting near-infrared fluorescence diagnosis and photothermal therapy of echinococcosis. The biomimetic Lipo-ICG theranostic agents exhibited excellent active-targeting ability and good photothermal therapy ability. Under the near-infrared fluorescence imaging-guided, the mice infected by echinococcosis were cured. It is the first prospective study using biomimetic liposomes for the diagnosis and treatment of zoonotic diseases, providing a promising method to explore functional nanomaterials for rapid and efficient phototheranostics of infectious diseases in the underdeveloped area.

The manuscript can be accepted after a minor revision.

  1. The authors mentioned the particle size of Lipo-ICG decreased from ~128 nm to ~119 nm after the cell membrane camouflaging treatment. It suggested to explain this phenomenon.
  2. The abbreviation of the materials should be uniformed in the text and the figures.
  3. Why the authors chose neutrophil membrane to modify the Lipo-ICG?
  4. As compared with other biomimetic nanoprobes, what’re the advantages of materials modified by the neutrophil cell membrane?
  5. The full name of the abbreviations should be given such as FBS, EDTA, and so on.
  6. Does the fluorescence intensity of ICG change after encapsulation in the liposome?
  7. The subscript of DSPE-PEG2000 at line 69 of P2 is not in conformity with its subscript at line 74 of P2. The subscript of CHCl3 isn’t correct.
  8. At line 131 of P4, “Absoprtion spectra (e) and (f) emission fluorescence spectra” need to be revised as “Absoprtion spectra (e) and emission fluorescence spectra (f)”.
  9. At line 171 of P5, “kidneys” should be changed to “kidney”.
  10. At line 186 of P6, “which was higher than that of Lipo-ICG-treated group” should be changed to “which were higher than that of Lipo-ICG-treated group.”
  11. At line 193 of P6, “produce” should be changed to “produced”.
  12. Related references Exploration 2021, 3, 20210089; Adv. Mater. 2022, 34, 2102797; Am. Chem. Soc. 2020, 142, 10383. should be cited in the revised manuscript.

Reviewer 4 Report

In this manuscript, Liu and co-workers are reporting an interesting indocyanine green incoporated lipososme particles for phototheranostics application. The findings of this research is interesting to the bioimaging community. However the current format of the representation of the manuscript is bit complicated and I believe authors can simplify some of the regions better. I would highly recommend authors to use schemes and figures to briefly explain the mechanism of these theranostics agents which will be helpful for the readers. In addition, authors can also provide a summary of previously reported theranostic agents in brief with references to highlight the importance of this research. In addition, I would also like to recommend following modifications:

(1).  What is the logic behind using the two lipid types authors choose for this liposome formation?

(2). There are some textual errors such as writing CHCl3, which must be corrected.

(3). In figure 1, what is the excitation wavelength to obtain fluorescence spectra? Can authors calculate the molar absorptivity coefficient from the absorbance data?

(4). Figure 2a and b must provide error bar to strengthen the scientific soundness.

(5). The experimental set up associated with figure 3 has not been explain clearly. This is a critically important set of experimental data which authors should describe and discuss in detail.

(6). In figure 1d in bar chats, what is the confidence interval? this is not mentioned in the figure. Also what is the excitation wavelength for the imaging in this figure?

(7). The conclusion of the manuscript can be further improved to highlight the importance of the experimental findings. The current version does not conclude the findings very well.

Reviewer 5 Report

The authors have synthesized a phototheranostic system using neutrophil membrane camouflaging Lipo-ICG for near-infrared photothermal therapy of echinococcosis in mouse liver. The story is relatively comprehensive, engaging, and innovative, and I suggest accepting the work after minor revisions, as indicated below.

Major concerns:
(1) The background of this work can be expanded. I suggest authors discuss relative papers in detail and compare them (pros and cons) with the phototheranostic system demonstrated in this paper.
(2) Clinical translation would be the ultimate goal of Neu-lipo-ICG. I understand the toxicity is low, but what would be the next step for FDA approval? I know many other similar contrast agents composed of bio-safe materials waiting for approval. What is the hurdle? Please discuss.
(3) Liver is an optically absorptive organ due to the high blood volume, even for the NIR wavelengths. Please quantify the effective depth of phototherapy in the liver if possible. Again, how to improve this for clinical translation (e.g., microwave)?

Minor concerns:
(4) Please check the unit for power density.
(5) In line 144, how would you define “efficient”?
(6) Any safety limit for laser radiation in phototherapy?
(7) Why do Neu-lipo-ICG particles look smaller than Lipo-ICG in Fig. 1b?

Round 2

Reviewer 1 Report

The authors have addressed most of my previous concerns, and the quality of the manuscript has been improved. I believe it can be accepted for publication.